# Vagus Nerve Stimulation Modulates Inflammation in Treatment-Resistant Depression Patients: A Pilot Study

**DOI:** 10.3390/ijms25052679

**Published:** 2024-02-26

**Authors:** Paul Lespérance, Véronique Desbeaumes Jodoin, David Drouin, Frédéric Racicot, Jean-Philippe Miron, Christophe Longpré-Poirier, Marie-Pierre Fournier-Gosselin, Paméla Thebault, Réjean Lapointe, Nathalie Arbour, Jean-François Cailhier

**Affiliations:** 1Department of Psychiatry, Centre Hospitalier de l’Université de Montréal (CHUM), Université de Montréal, Montreal, QC H2X 0C1, Canada; 2Department of Medicine, Université de Montréal, Montreal, QC H3T 1J4, Canada; 3Division of Neurosurgery, CHUM, Université de Montréal, Montreal, QC H2X 0C1, Canada; 4Centre de Recherche du CHUM (CRCHUM), Institut du Cancer de Montréal, Montreal, QC H2X 0A9, Canada; 5Department of Neurosciences, Université de Montréal and CRCHUM, Montreal, QC H2X 0A9, Canada; 6Department of Medicine, Renal Division, CHUM, Université de Montréal, Montreal, QC H2X 0C1, Canada

**Keywords:** neuromodulation, major depression, inflammation, blood–brain barrier

## Abstract

Vagal neurostimulation (VNS) is used for the treatment of epilepsy and major medical-refractory depression. VNS has neuropsychiatric functions and systemic anti-inflammatory activity. The objective of this study is to measure the clinical efficacy and impact of VNS modulation in depressive patients. Six patients with refractory depression were enrolled. Depression symptoms were assessed with the Montgomery–Asberg Depression Rating, and anxiety symptoms with the Hamilton Anxiety Rating Scale. Plasmas were harvested prospectively before the implantation of VNS (baseline) and up to 4 years or more after continuous therapy. Forty soluble molecules were measured in the plasma by multiplex assays. Following VNS, the reduction in the mean depression severity score was 59.9% and the response rate was 87%. Anxiety levels were also greatly reduced. IL-7, CXCL8, CCL2, CCL13, CCL17, CCL22, Flt-1 and VEGFc levels were significantly lowered, whereas bFGF levels were increased (*p* values ranging from 0.004 to 0.02). This exploratory study is the first to focus on the long-term efficacy of VNS and its consequences on inflammatory biomarkers. VNS may modulate inflammation via an increase in blood–brain barrier integrity and a reduction in inflammatory cell recruitment. This opens the door to new pathways involved in the treatment of refractory depression.

## 1. Introduction

Recent hypotheses on the etiopathogenesis of major depressive disorder have proposed that proinflammatory cytokines may play an important part in the onset and persistence of depressive symptoms. Studies report that the plasma levels of proinflammatory cytokines such as interleukin-1β (IL-1β), IL-6, IL-12, CC chemokine Ligand (CCL-2), Tumor Necrosis Factor (TNF)-α, prostaglandin E2 are increased in patients suffering from depression [1]. Vagus nerve stimulation (VNS) is an approved adjunctive therapy for treatment-resistant depression (TRD). The vagus nerve is composed of afferent fibers sending sensory information to the brain (80%) and efferent fibers relaying data from the brain back into the body (20%). It is also closely related to the cortical–limbic–thalamic–striatal neural circuit involved in emotional and cognitive functions. Although its exact mechanisms remain unclear, VNS is thought to influence microglial cells directly or indirectly through complex brain-immune system interactions [2]. It was shown that VNS influences neurotransmitters implicated in mood disorders, such as serotonin and norepinephrine, and increases the brain-derived neurotrophic factor (BDNF), also increased by pharmacological antidepressants. Interestingly, recent reports have also suggested that VNS has an impact on systemic inflammation via the modulation of cytokines (IL-1β, IL-6, and TNFα) [3], such as in inflammatory bowel diseases. Furthermore, the vagus nerve efferent pathway also influences immune cells in the spleen, generating an immunosuppressive environment [3]. This new rationale led to a better understanding of approved therapies for TRD. Many reports demonstrated changes in the inflammatory proteins measured in the blood of depressive patients after pharmacological treatment, such as chemokines relevant to inflammatory cell recruitment, proteins relevant to blood–brain barrier integrity, and acute inflammatory cytokines. Although there is a growing interest in the study of inflammation in depressive patients, very few have examined the inflammatory biomarkers in treatment-resistant depressive patients. To our knowledge, no study has explored whether VNS treatment has induced inflammatory changes in TRD patients.

In this limited and exploratory study, we measured the clinical efficacy of VNS treatment in TRD and inflammatory proteins at the time of implantation and later in the course of the disease. Here, despite a low number of patients, we described significant modulations of several inflammatory proteins after more than 4 years of continuous VNS treatment, suggesting a modulation of the inflammation in TRD.

## 2. Results

### 2.1. VNS Induced a Significant and Sustained Clinical Response in TRD Patients

Table 1 represents clinical characteristics at baseline of our patients. Appendix A includes all the other clinical, biological characteristics and VNS settings of the patients. The mean baseline MADRS score was 24.5 (±7.2), representing a moderate level of depression. Duration of the current depressive episodes at the time of VNS surgery was, on average, 43 months (18–72 months). At the time of the last blood harvest, we observed a 76.3% drop in the mean MADRS score, with 5 out of 6 patients achieving a clinical response (MADRS reduction of at least 50%). Anxiety symptoms were also reduced significantly (59.9%) (see Table 2).

### 2.2. VNS Modulates Inflammatory Proteins in Sera of TRD Patients

Amongst the 40 molecules measured, we found significant changes between the pre-implantation and post-implantation levels of several chemokines/pro-inflammatory cytokines and proteins. VNS stimulation reduced the levels of IL-7, CXCL8, CCL2, CCL13, CCL17, CCL22, Flt-1 and VEGF-C, whereas it increased bFGF levels (Table 3 for patients’ and healthy donors’ values and Figure 1 for individualized variations). Interestingly, we did not observe any significant variation in the levels of inflammatory proteins previously reported in depressive patients, namely TNF-α, IL-6 and IFNγ (Figure 2). Moreover, IL-1β levels were undetectable. All the other measured molecules not presented here did not variate significantly before or after treatment. We found no correlation between the variation of inflammatory proteins in the plasma and changes in the MADRS score.

## 3. Discussion

We measured inflammatory proteins in the plasma of TRD patients treated with VNS and we highlighted significant changes before and after the implantation. VNS treatment resulted in a significant modulation of inflammation by reducing pro-inflammatory cytokine and chemokine plasma levels. IL-7, CXCL8, CCL2, CCL13, CCL17 and CCL22 were reduced after more than 4 years of VNS treatment. Interestingly, these inflammatory mediators are different from the ones previously reported in depression (TNF-α, IL-1β and IL-6). The latter are usually involved in acute inflammation and they were studied in patients with major depression, responding or not responding to treatment. Since we recruited only treated TRD patients, they may suffer from a more chronic illness, explaining why we did not observe changes after VNS implantation.

IL-7 is crucial for T cell homeostasis, whereas CXCL8 is a chemokine crucial to promote neutrophil recruitment. Both cytokines were found to be elevated in depressive patients and reduced to the levels of healthy controls after pharmacologic treatment. Interestingly, we observed the same effect on IL-7 and CXCL8 after VNS therapy, suggesting that anti-depressive therapies such as drugs and VNS may exert their effects partly through a reduction in these pro-inflammatory mediators [4].

We also found that CCL2, CCL13 and CCL17 levels were reduced with VNS therapy. These chemokines have been shown to promote immune cell infiltration in various central nervous system diseases [5], such as multiple sclerosis [6]. High levels of CCL2 and CCL13 were observed in patients with depression compared to healthy control, evoking the possibility that immune cell mobilization may be involved in the disease [7]. Thus, VNS is likely to act as a modulator of inflammation in treatment-resistant major depression patients via leukocyte recruitment in the brain.

CCL22 is a chemokine involved in the pathophysiology of infectious and neoplastic diseases. One recent study suggested that CCL22 could be used as a marker of treatment response, with levels increasing six weeks after the beginning of pharmacological therapy. Increased levels of CCL22 were also initially linked to better responsiveness to pharmacological treatment [8]. In our study, we found a decrease in CCL22 with VNS therapy. Since we studied inflammatory protein levels after a few years of VNS treatment, this suggests that CCL22 may have a transitory role in promoting response to anti-depressive treatment.

VEGF-C and sFlt-1 levels also diminished, whereas bFGF levels increased with VNS. The blood–brain barrier is thought to play a crucial role in major depression patients [9]. Several molecules maintain their integrity and homeostasis. VEGF A and C promote blood–brain barrier permeability, whereas sFlt-1 plays a role in counteracting this effect [10] and bFGF plays a role in promoting blood–brain barrier integrity [9]. The role of VEGF in major depressive disorder is still ambiguous, but low levels are associated with greater vulnerability to stress, thus a higher risk of developing depressive symptoms. In counterpart, low levels are also associated with a better response to pharmacological treatment. Studies show contraindicatory reports regarding VEGF levels in depression [11]. This disparity may be explained by the variations in the severity/chronicity of the disease and the conditions leading to the development of depression. On the other hand, bFGF levels increased with VNS treatment. Previous studies have linked depression with lower levels of circulating bFGF [12]. VNS may play a role in the recovery of blood–brain barrier integrity via bFGF and sFlt-1 modulation. Based on our observations, we suggest that VNS promote the recovery of blood–brain barrier integrity by reducing the VEGF-C regulator, leading to a compensatory reduction in sFlt-1 levels and an increase in bFGF levels. This would result in a decrease in blood–brain barrier permeability and brain infiltration of leukocytes.

The efficacy of VNS therapy in our group of 6 patients was high, and above the rates reported in the literature. Yet, a recent 5-year longitudinal registry study of almost 500 patients did find strong evidence for long-term benefits of VNS therapy on over 300 patients in the treatment-as-usual (TAU) group, and a lower relapse rate, rekindling interest in this technology in TRD [13].

Obviously, this represents a pilot study with heterogeneity in sample collections after VNS implantation, and very few patients. Despite these limitations, we highlighted statistically significant differences between the levels of various inflammatory markers.

Our patients in this exploratory study also show prolonged VNS anti-depressive effects, associated with a modulation of inflammation via several mechanisms. To our knowledge, this is the first study demonstrating the possible impact of VNS on inflammatory proteins related to blood–brain barrier integrity and inflammatory cell recruitment in the brain. However, because of several limitations in the current work (small sample size, referral bias, no TRD control group), further studies will be needed to better characterize these changes. First, the small sample size limited us to detect only large effects on plasma proteins. Smaller variations may have been missed due to a lack of statistical power. It was also impossible to define clear cut-off levels characterizing patients versus normal donors. Statistical analyses of non-responders versus responders could not be performed due to the low number of patients. Also, we could not show a significant correlation between the MADRS score and inflammation because of insufficient statistical power. A larger cohort would be needed to determine if baseline inflammatory protein levels (low/high) are predictive of response to therapy, and to find out if modulation of other inflammation markers is in play with this treatment.

## 4. Materials and Methods

### 4.1. Patients

This study was approved by the local ethics committee and performed in accordance with the 3 councils declaration. All patients provided written informed consent. Six patients with TRD, mean age at implantation (48.8 ± 5.8; range 41–57), who underwent standard neurosurgical VNS implantation (Cyberonics Model 102 pulse generator, LivaNova, Boston, MA, USA) at the Centre Hospitalier de l’Université de Montréal (CHUM) between 2007 and 2010 (mean age at implantation; 48.8 ± 5.8) were recruited. No complications were observed in our patients. Inclusion criteria were previously described, grossly defined as unipolar or bipolar disorder with a partial response or no response to at least 4 antidepressant medications. Exclusion criteria included history of auto-immune disease, treatment with immunomodulatory medication, ongoing infection, spontaneous remission before surgery, other psychiatric conditions, and medical contraindication, such as clinically relevant cardiovascular disease, active cancer, and pregnancy [14]. All patients were assessed using the Montgomery–Asberg Depression Rating Scale (MADRS) and Hamilton Anxiety Rating Scale (HAM-A) to measure depressive and anxious symptoms at baseline, 12 months, 24 months, and at the time of the second blood harvest. Response to therapy in depression was defined as a reduction of at least 50% on the MADRS. Blood samples were obtained at 8AM before implantation (baseline), and up to 4 years or more after continuous VNS therapy (range 56–93 months). We also obtained blood samples from 6 healthy donors. EDTA blood was centrifuged and stored at −80 °C.

### 4.2. Measurement of Inflammatory Proteins in Plasma

Plasma soluble inflammatory proteins were analyzed by multiplex (high sensitivity V-PLEX Human Biomarker 40-Plex Kit from MesoScale Discovery, Rockville, MD, USA). The analytes measured were C reactive protein, CCL11 (Eotaxin), CCL26 (Eotaxin-3), basic Fibroblast growth Factor (bFGF), Granulocyte-macrophage colony-stimulating factor (GM-CSF), Intercellular Adhesion Molecule (ICAM-1), Interferon (IFN)-γ, IL-10, IL-12/IL-23p40, IL-12p70, IL-13, IL-15, IL-16, IL-17A, IL-1α, IL-1β, IL-2, IL-4, IL-5, IL-6, IL-7, C-X-C Motif Chemokine Ligand (CXCL)-8, CXCL10, CCL2, CCL3, CCL4, CCL13, CCL22, Placental growth factor (PlGF), Serum Amyloid A, CCL17, Tyrosine kinase 2 (Tie)-2, TNF-α, TNF-β, Intercellular Adhesion Molecule (VCAM)-1, Vascular Endothelial growth factor (VEGF)-A, VEGF-C, VEGF-D, VEGF Receptor or Fms-like tyrosine kinase (Flt-1).

### 4.3. Statistics

Analyses were performed using IBM SPSS Statistics version 25 (IBM Corporation, Amonk, NY, USA). Data obtained from our 8 patients were analyzed using descriptive statistics and repeated measures multivariate ANOVAs (MANOVA), as Pearson correlations revealed several variations in inflammatory proteins. Differences with *p* < 0.05 were considered to be statistically significant.

## 5. Conclusions

Here, we suggest that VNS therapy is associated with modulation of the immune system, inflammation, and blood–brain barrier function. Several chemokines were decreased in the plasma of TRD patients with ongoing VNS therapy for at least 4 years, such as CCL2, CCL13 and CCL17 associated with leukocyte recruitment. Proteins related to blood–brain barrier homeostasis, integrity, and permeability were also modulated (VEGF, sFlt1 and bFGF). Thus, we believe that VNS may reestablish blood–brain barrier integrity and reduce recruitment of inflammatory cells in the brain. This could lead to improvements in depressive symptoms or remission. This study provides pilot data that may warrant and compel future consortia to further test this hypothesis.

## Figures and Tables

**Figure 1 ijms-25-02679-f001:**
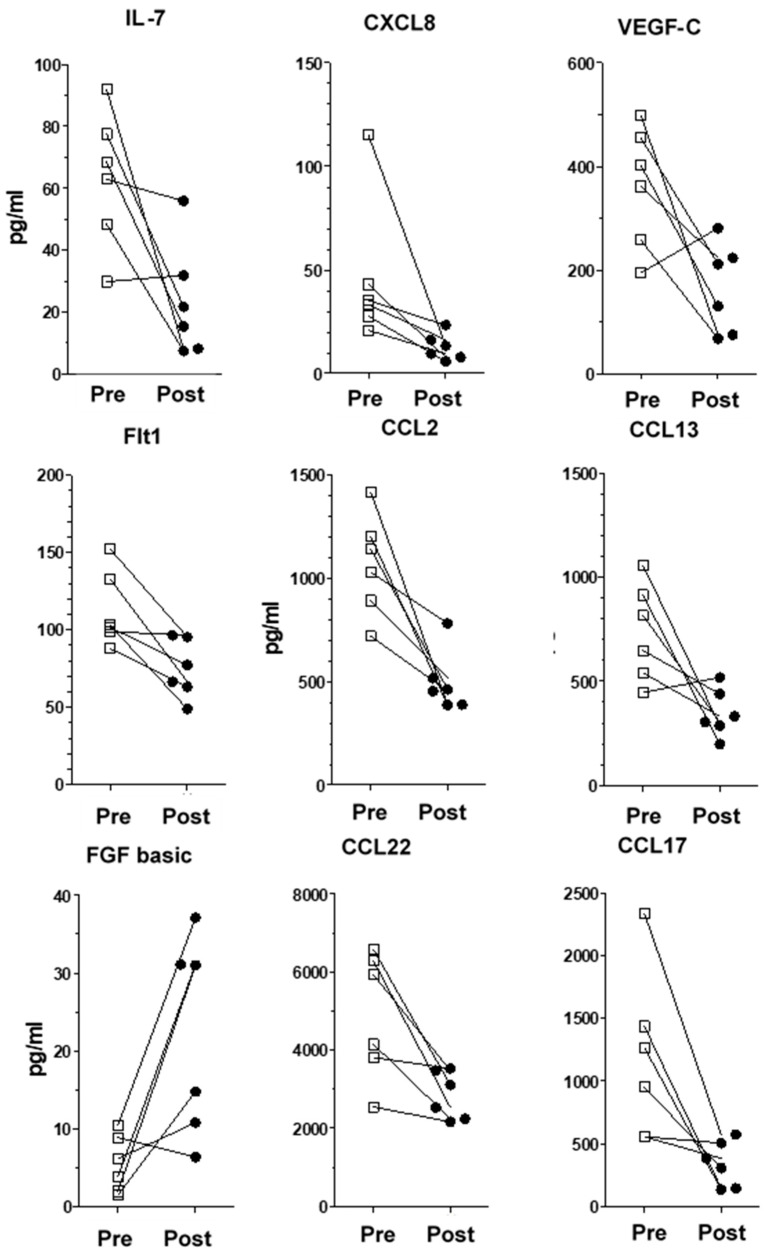
Individualized variations and dispersion of cytokine/chemokine levels for each VNS patient. Pre-implantation and post-implantation values for IL-7, CXCL8, VEGF-C, Flt-1, CCL2, CCL13, bFGF, CCL17 and CCL22 and bFGF (values in pg/mL). Squares are pre-treatment values, black circles are post-treament values.

**Figure 2 ijms-25-02679-f002:**
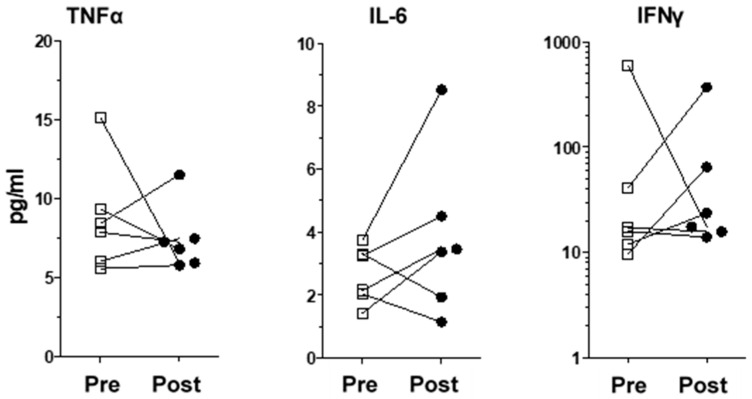
Levels of inflammatory proteins previously reported to be modulated in depressive patients. Individualized variations for TNF-α, IL-6 and IFNγ in VNS patients were not significant. Squares are pre-treatment values, black circles are post-treament values.

**Table 1 ijms-25-02679-t001:** Characteristics and baseline clinical evaluation of patients treated with VNS. MADRS = Montgomery–Asberg Depression Rating Scale, 0 to 6—normal/symptom absent, 7 to 19—mild depression, 20 to 34—moderate depression, >34—severe depression; HAM-A = Hamilton Anxiety Rating Scale, 0 to 17—mild anxiety, 18 to 24—mild or moderate anxiety, 11 to 14—moderate anxiety, 25 to 30—severe anxiety.

Characteristics	Patients (*n* = 6)	Healthy Donors (*n* = 6)
*Age*	48.83 (41.0–57.0)	32.67 (24–51)
*Sex*		
*Men*	2 (33.3%)	2 (33.3%)
*Women*	4 (66.6%)	4 (66.6%)
*Initial diagnosis*		-
*Bipolar disorder*	2 (33.33%)	
*Unipolar disorder*	4 (66.66%)	
*Baseline clinical evaluation*		-
*MADRS score*	24.5 (14–34)	
*HAM-A score*	16.2 (9–29)	

**Table 2 ijms-25-02679-t002:** Depression and Anxiety scores of patients treated with VNS.

	Baseline Score	Post-Treatment Score	Change (%)	*p*
*MADRS Score*	24.5 (14–34)	5.8 (1–8)	76.3	0.00074
*HAM-A Score*	16.2 (9–29)	6.5 (2–13)	59.9	0.0031

**Table 3 ijms-25-02679-t003:** Cytokine and chemokine levels in healthy controls and patients treated with VNS (baseline and post-treatment) with quantified difference. *p*-value represents the mean difference at baseline and after treatment, () for standard deviation.

Cytokine/Chemokine	Healthy Controls (pg/mL)	Baseline (pg/mL)	Post-Treatment (pg/mL)	Change (%)	*p*
*IL-7*	4.0 (0.97)	63.2 (21.9)	23.4 (18.4)	−63.0	0.029
*CXCL8*	2.0 (0.5)	40.5 (31.8)	5.8 (2.0)	−85.7	0.041
*CCL2*	40.0 (11.2)	1065.2 (244.3)	498.2 (146.5)	−53.2	0.007
*CCL13*	63.0 (22.6)	736.4 (232.6)	346.3 (114.3)	−53.0	0.034
*CCL17*	100.0 (25.0)	1182.4 (669.7)	340.1 (181.9)	−71.2	0.028
*CCL22*	1162.0 (265.0)	4882.8 (1615.7)	2843.4 (610.4)	−41.8	0.020
*Flt-1*	40.0 (5.0)	112.7 (24.5)	74.5 (18.8)	−32.8	0.013
*VEGF-C*	37.0 (25.0)	361.9 (116.6)	165.2 (86.8)	−54.4	0.036
*bFGF*	39.0 (42.0)	5.5 (3.7)	21.9 (12.7)	298.2	0.004

## Data Availability

Details on data supporting the results presented here are available upon reasonable request to the corresponding author. The data are not publicly available due to [lack of plateform availability and funding].

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
