# Peer review of "Vagus Nerve Stimulation Modulates Inflammation in Treatment-Resistant Depression Patients: A Pilot Study"

_ijms, 2024, doi:10.3390/ijms25052679_

Round 1
Reviewer 1 Report
Comments and Suggestions for Authors
Dear Paul Lesperance and co-authors,
Your pilot study is very encouraging and it is important to report it since it supports Vagus Nerve Stimulation as an alternative and effective treatment of depression. I recommend to modify the title , incorporating "Preliminary data" or "Pilot data" as all study was done only on 6 patients. The second recommendation is to add more details into Methods regarding the implantation of Vagus Nerve Stimulator (Cyberonics Model 102 pulse generator), or if it is standard neurosurgery , please, add reference for it.
Overwise, congratulations with your exciting results!
Author Response
We thank the reviewer for the kind comments.
We have modified our title as suggested and have specified how the neurostimulators were installed.
Reviewer 2 Report
Comments and Suggestions for Authors
In this communication, the authors examine the clinical efficacy of VNS modulation in six depressive patients. Depression symptoms were assessed with specific clinical scale, and plasma were harvested prospectively before implantation of VNS (baseline) and up to after four years or more of continuous therapy, measuring 40 soluble molecules by multiplex assay. Authors report an improvement in: reduction of the mean depression severity score, decrease in IL-7, CXCL8, CCL2, CCL13, CCL17, CCL22, Flt-1 and increase in bFGF levels.
They conclude that VNS may modulate inflammation via an increase in the blood-brain barrier integrity and a reduction in inflammatory cell recruitment.
Main question:
- Specify the characteristics of the patients:
Supplementation? antioxidants? smoke? height and weight of the patients? specific diets? years from diagnosis, and from surgery?
It is recommended to make a table indicating the clinical/pharmacological characteristics for each individual subject from diagnosis to surgery, and the timing of blood sampling, given that not all were drawn at 1 year (implement table 3).
- Insert the part of Vagal neurostimulation (VNS): device, implantation, complication after surgery, electrical impulse, setting parameters of stimulation, adjustment device and parameters over time
- Given that there are very few patients, with heterogeneity in terms of sample collection, a limitations paragraph is recommended
- Table 2 and figure 1 give the same message. This is redundant, it is recommended to delete table or figure (significant features are missing in the figure).
- Is the biological effect related to stimulation time/parameters?
Review the introduction and discussion from this perspective
Finally, insert an experimental scheme.
Comments on the Quality of English Language
None
Author Response
We thank the reviewer for the comments.
Here is our reply:
Main question:
Specify the characteristics of the patients:
- As suggested, we will include a supplementary table 1 that will highlight the requested clinical characteristics. We will include it there for graphical reason, since it is larger than the usual format. It is attached to this reply.
- Given that there are very few patients, with heterogeneity in terms of sample collection, a limitations paragraph is recommended
- We have included such a paragraph and sentences in the manuscript.
- Table 2 and figure 1 give the same message. This is redundant, it is recommended to delete table or figure (significant features are missing in the figure).
- Table 2 included values from normal subjects that are absent from figure 1. Therefore, we will leave both elements in the manuscript.
- Is the biological effect related to stimulation time/parameters?
- We do not think so.

Round 2
Reviewer 2 Report
Comments and Suggestions for Authors
Dear authors,
Table 3 and figure 1 give the same message. This is redundant, it is recommended to delete table or figure.
Add Sample size in "Statistics" paragraph.
Best regards
Author Response
We have modified the statistics paragraph to include our sample size. We think that table 3 and figure 1 represent different data, values in normal subject and individualized data for relevant cytokines for each patients. Therefore, we think that both need to be included in our manuscript.